

# Effects of different proportions of organic substitution for mineral fertilizers on soil methanogenic and methanotrophic communities in paddy fields

Dandan Yuan[1,2], Keke Dang[1,2,3], Jing Yin[1,2], Han Liu[4], Tingting Ma[1,2], Jia Liu[4] and Xingjia Xiang[1,2,3]

[1] School of Resources and Environmental Engineering, Anhui University, He Fei, Anhui Province, China
[2] Anhui Province Key Laboratory of Wetland Ecosystem Protection and Restoration, He Fei, Anhui Province, China
[3] Anhui Shengjin Lake Wetland Ecology National Long-term Scientific Research Base, He Fei, Anhui Province, China
[4] Soil and Fertilizer & Resources and Environment Institute, Jiangxi Academy of Agricultural Sciences, Nanchang, Jiangxi Province, China

Corresponding authors
Jia Liu, liujia422@126.com
Xingjia Xiang, xjxiang@ahu.edu.cn

## ABSTRACT

Mineral fertilizers are widely used to improve rice yields, but their overuse has caused severe environmental problems. Replacing mineral fertilizers with organic alternatives might be an effective practice for enhancing agro-ecosystems. This study investigated treatments with varying proportions of organic substitution to determine the optimal approach for increasing soil fertility and rice yield. In addition, the relationship between soil methane emission characteristics and associated microbial communities was studied by microcosm experiments and high-throughput sequencing to assess greenhouse gas emissions. Compared with mineral fertilizers alone, treatment with organic substitution, especially at high proportions, increased soil pH, fertility, and crop yield. Treatment with a medium proportion of organic substitution increased cumulative methane ($CH_4$) emissions by 44.8% relative to mineral fertilization alone, but that with low and high proportions showed similar emissions compared with mineral fertilization alone. Organic substitution treatment significantly increased the gene copy numbers of soil methanogens and methanotrophs, with the highest increases observed under high proportions of organic substitution. The gene copy number of methanogens increased by 4.87 times, and that of methanophiles increased by 13.11 times. Additionally, organic substitution treatment significantly changed their community compositions. High organic substitution was associated with an exceptionally high abundance of methanotrophs. Treatment with a high proportion of organic substitution enhanced the relative abundance of Type I taxa of methanotrophs and increased soil pH to trigger higher *pmoA* abundance, thus strengthening methane oxidation capacity without additional cumulative $CH_4$ emissions compared with mineral fertilizers alone. Besides, treatment with a high proportion of organic substitution increased crop yield and reduced the amount of mineral fertilizers needed, resulting in less environmental pollution. This study comprehensively evaluated the effects of organic substitution for mineral fertilizers, providing an essential theoretical basis for the sustainable development of agriculture.

## INTRODUCTION

Fertilization is essential for maintaining high rice yields. Although mineral fertilizers are widely used to enrich soils and boost crop yields (*Guo, Liu & He, 2022*), their overuse can lead to soil biodiversity loss, soil acidification, and environmental pollution (*Liu & Diamond, 2005*; *Zhu, Xiong & Xing, 2005*). Organic fertilizers, such as pig manure and Chinese milk vetch (*Astragalus sinicus* L.), effectively improve soil fertility and sustain crop productivity with less environmental risk compared with mineral fertilizers alone (*Yu et al., 2020a*; *Zeng & Li, 2022*). Previous studies have demonstrated that combining organic and mineral fertilizers can effectively improve soil fertility by increasing soil organic carbon and nitrogen contents and enhancing the activity of soil microorganisms. This combination leads to a higher fertilizer utilization rate and reduced environmental pollution, along with improved crop yields (*Hou et al., 2012*; *Pan et al., 2009*). However, these studies typically applied organic fertilizers on top of mineral fertilizers without exploring the partial replacement of mineral fertilizers with organic alternatives. Thus, systematic research on the response of agro-ecosystems to varying proportions of organic substitution for mineral fertilizers in paddy fields is lacking.

Different proportions of organic substitution for mineral fertilizers impact greenhouse gas emissions differently. Methane ($CH_4$) is the second most significant greenhouse gas globally, with a global warming potential (GWP) 25 times greater than that of $CO_2$ (*Forster et al., 2008*). Rice is the largest food crop in China, and its yield is crucial for food security and social stability. Rice paddies are a major source of methane emissions, releasing 25–300 Tg of methane gas annually (*Bridgham et al., 2013*), accounting for 9% of human-caused $CH_4$ emissions (*Masson-Delmotte et al., 2021*). Reducing methane emissions from paddy ecosystems is essential for sustainable environmental development.

Methane emission fluxes in soil are influenced by the balance between methanogens and methanotrophs. The emission of methane and its associated microbial communities has been key topics of research across various ecosystems. A previous study found that introducing *Sonneratia apetala* into mangrove ecosystems altered the methane-cycling microbial community, increasing methane emissions (*Yu et al., 2020b*). *Waldo et al. (2022)* studied the effects of wetland plants on methane metabolism, revealing the significant role that mature plants play in affecting the methanotrophs in their rhizosphere. Methane is mainly produced by acetotrophic and hydrogenotrophic methanogens. The methanotrophs include Type I and Type II methanotrophs (*Hanson & Hanson, 1996*; *Trotsenko & Murrell, 2008*). The application of different fertilizers to soil affected the structure of methanogenic and methanotrophic communities differently. The long-term application of manure increased the gene copy number of *mcrA* but had little effect on the *pmoA* gene copy number (*Zhang et al., 2018*). Manure amendments increased the abundance of methanogens and methanotrophs but suppressed Type I methanotrophs in rice paddies (*Wang et al., 2020*).

The combined application of leguminous green manure and straw significantly increased the relative abundance of *Methanosarcinales* compared with mineral fertilizer alone (*Zhou et al., 2020a*). Although many previous studies have examined the effects of organic and mineral fertilizers on methane-associated microbial communities, the link between soil methane emission characteristics and the associated microbial communities following different proportions of organic substitution for mineral fertilizers is not well understood.

This study investigated the effects of different proportions of organic substitution for mineral fertilizers on soil fertility, crop yield, and methane-associated microbial community structures at a double-season rice fertilization experiment station. The study aimed to address two scientific questions: (1) What are the effects of different proportions of organic substitution for mineral fertilizers on soil fertility and crop yields? (2) How do different proportions of organic substitution for mineral fertilizers affect methane emission characteristics and the structures of methanogenic and methanotrophic communities in paddy soil?

## MATERIALS AND METHODS

### Site description

The long-term experiment, initiated in 1984, was conducted at the Institute of Soil Fertilizer and Resource Environment, Jiangxi Academy of Agricultural Sciences, Nanchang, Jiangxi Province (28°33′92″N, 115°56′25″E; 25 m). The average annual temperature and rainfall were 17.5 °C and 1,600 mm, respectively. The soil at the test site was a medium-retention yellow mud field developed from a Quaternary sub-red clay matrix, classified as Chromic Alisols (Cutanic, Dystric, and Loamic) according to the World Reference Base for Soil Resources (WRB) in 2022 (*IUSS Working Group WRB, 2022*). The cropping system consisted of double-season rice (early rice–late rice). The basic chemical properties of the soil from 0 to 20 cm depth before the experiment in 1984 were as follows: pH 6.5, organic matter 25.6 g/kg, total nitrogen 1.36 g/kg, total phosphorus 0.49 g/kg, alkaline hydrolyzable nitrogen 81.6 mg/kg, available phosphorus 20.8 mg/kg, and available potassium 35.0 mg/kg.

### Experimental design

The experiment was designed as random blocks, with each plot covering 33.3 m$^2$ (11.1 × 3 m$^2$) and an interval of 0.50 m between plots. The plots were separated by a 0.50-m-wide cement ridge, and each plot was independently drained and irrigated. The experiment included five treatments, each with nine replicates: no fertilizer (CK), 100% mineral nitrogen, phosphorus, and potassium (NPK) fertilizers, 70% mineral fertilizer and 30% organic fertilizer (M30), 50% mineral fertilizer and 50% organic fertilizer (M50), and 30% mineral fertilizer and 70% organic fertilizer (M70). For early rice, the 100% mineral fertilizer was applied with pure N at 150 kg/ha, P$_2$O$_5$ at 60 kg/ha, and K$_2$O at 150 kg/ha. For late rice, it was applied with pure N at 180 kg/ha, P$_2$O$_5$ at 60 kg/ha, and K$_2$O at 150 kg/ha (Table S1). The organic fertilizer for early rice was Chinese milk vetch (*A. sinicus* L.), containing N at 3.03 g/kg, P$_2$O$_5$ at 0.8 g/kg, and K$_2$O at 2.3 g/kg. The organic fertilizer for late rice was rotted pig manure, containing N at 4.5 g/kg, P$_2$O$_5$ at 1.9 g/kg, and K$_2$O at 6.0 g/kg. The treatments (except CK) were designed to have equal nutrient quantities

based on nitrogen quantity, with mineral fertilizers filling any phosphorus and potassium shortfalls. Phosphorus and organic fertilizers were applied as basal fertilizers. Notably, 50% of nitrogen fertilizer was used as basal fertilizers (Early Rice: April 20; Late Rice: July 18), 25% in the tillering stage (Early: May 26; Late rice: August 20), and 25% in the spike stage (Early Rice: June 10; Late Rice: September 9). Further, 50% of potassium was used in the tillering stage and 50% in the spike stage.

## Sample collection and analysis

Early rice was transplanted in late April and harvested in mid-July, whereas late rice was transplanted in late July and harvested in late October. Both early and late rice were collected in the maturity stage, and their yields were measured. Soil samples were collected on October 15, 2021. A 3-cm auger was used to collect soil from 0 to 20 cm depth at four corners and the center of each plot, and these samples were mixed to create one composite sample per plot. The collected soil was processed to remove debris such as stones and plants and subsequently transported to the laboratory within 12 h. Upon arrival, fresh soil samples were sieved through a 2-mm stainless steel sieve and divided into three parts: one for chemical property determination (see supporting information), another for the microcosm experiment, and the last part for microbial sequencing.

## Microcosm experiment

A 36-day microcosm experiment was conducted at Jiangxi Academy of Agricultural Sciences to investigate the impact of different treatments on methane emissions in paddy fields. The 200 g of freshly sieved soil was added to a 500-mL culture flask for each treatment, and then 200 mL of distilled water was added to simulate the flooded conditions of a rice field. The flask was then tightly sealed to maintain a closed environment. Nine replicates were set up for each treatment, and the incubator was maintained at a constant temperature of 25 °C throughout the experiment. A 25-mL gas sample was collected from each flask daily using a syringe, and methane levels were measured immediately using a gas chromatograph (Agilent 7890B; Agilent Technologies, La Jolla, CA, USA). The supplementary materials provide detailed information for this section.

## DNA extraction

Genomic DNA extraction from 0.5 g of fresh soil was performed using the FastDNA SPIN Kit for soil (MP Biomedicals, Irvine, CA, USA) following the manufacturer's protocols.

## Quantitative real-time polymerase chain reaction

The primers MLf/MLr and A189F/mb661R were used to quantify the *mcrA* gene and *pmoA* gene, respectively (*Bourne, McDonald & Murrell, 2001*; *Luton et al., 2002*). The primer sequence for *mcrA* was GGTGGTGTMGGATTCACACAR-TAYGCWACAGC/TTCATTGCRTAGTTWGGRTAGTT, and the primer sequence for *pmoA* was GGNGACTGGGACTTCTGG/CCGGMGCAACGTCYTTACC. The real-time polymerase chain reaction (PCR) mixture consisted of 16.5 μL of ChamQ SYBR Colour qPCR Master Mix (Vazyme Biotech Co., Ltd., Nanjing, China), 0.8 μL of the forward primer (5 μM), 0.8 μL of the reverse primer (5 μM), and 2 μL of template DNA (10 ng).

The qPCR amplification program for the *mcrA* and *pmoA* genes included an initial hold at 95 °C for 5 min, followed by 40 cycles of denaturation at 95 °C for 5 s, annealing at 55 °C for 30 s, and extension at 72 °C for 40 s.

## Illumina MiSeq sequencing

The aforementioned two primer sets were used to amplify *mcrA* and *pmoA* genes for sequencing methanogenic and methanotrophic communities. The PCR reaction system comprised 25 μL, including 4 μL of 5 × FastPfu buffer, 0.4 μL of TransStart FastPfu Polymerase (Transgen Biotech, Beijing, China), 2 μL of dNTPs (2.5 mM), 0.8 μL of forward primer (5 μM), 0.8 μL of reverse primer (5 μM), and 1 μL of template DNA. Then, the amplicons were extracted using a 2% agarose gel, purified following the AxyPrep DNA Gel Extraction Kit (Axygen Biosciences, Union City, CA, USA) instructions, and quantified using the QuantiFluor-ST system. The purified amplicons were used for paired-end sequencing (2 × 250) on the Illumina MiSeq platform. The reaction conditions for Illumina MiSeq sequencing of methanogens were initial denaturation at 94 °C for 3 min; 35 cycles of denaturation at 94 °C for 25 s, annealing at 55 °C for 45 s, and extension at 72 °C for 60 s, followed by a final extension at 72 °C for 5 min. The reaction conditions for Illumina MiSeq sequencing of methanotrophs were as follows: initial denaturation at 95 °C for 5 min; 35 cycles of denaturation at 92 °C for 1 min, annealing at 55 °C for 1.5 min, and extension at 72 °C for 60 s, followed by a final extension at 72 °C for 5 min.

## Bioinformatics analysis

The data analysis was performed using QIIME 1 (version 1.9) software. Sequences with poor quality (average quality score below 30) were excluded from the analysis. The optimized sequences were clustered into Operational Taxonomic Units (OTUs) at a 93% similarity level using Usearch for both methanogen and methanotroph datasets (*Lüke & Frenzel, 2011*; *Zhou et al., 2020a*). The chimera and singleton OTUs were eliminated from the dataset. The representative sequence for each OTU was identified using the FunGene database (*Fish et al., 2013*). Randomly selected subsets comprising 52,767 and 30,462 sequences (the lowest sequence read depth) per sample were used to compare the compositions and diversity of methanogenic and methanotrophic communities, respectively, to mitigate potential calculation errors arising from varying sequencing depths across samples.

## Statistical analysis

The one-way analysis of variance module in SPSS 20 software was used to analyze differences in soil chemical properties, total methane emissions, *mcrA* and *pmoA* abundance, and α diversity index across various treatments (Tukey HSD, $P < 0.05$). The Pearson correlation analysis assessed the correlation between methane emissions, microbial $\alpha$ diversity index, and soil chemical properties. Non-metric multidimensional scale analysis was performed to elucidate differences in methane-associated microbial communities across various treatments.

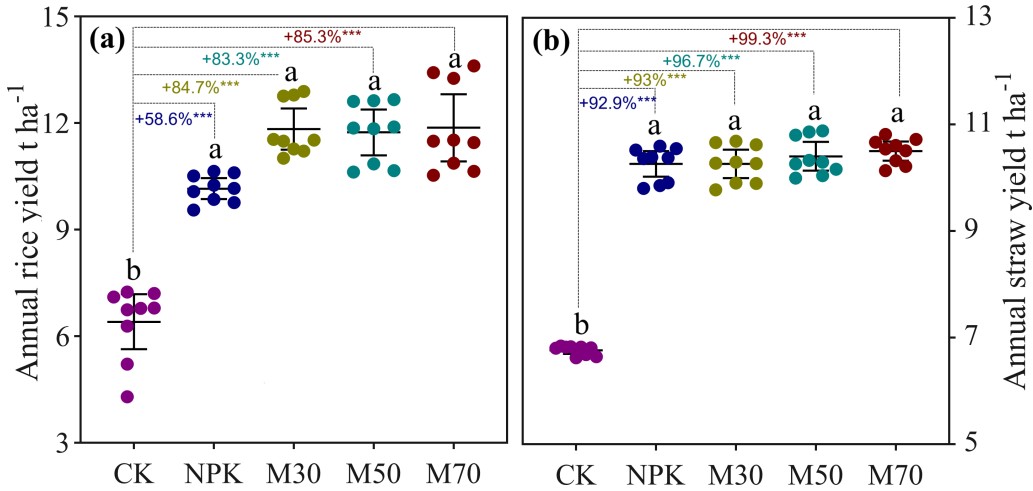

**Figure 1** **The grain and straw yield.** The total grain yield of early rice and late rice under different treatments (A); the total straw yield of early rice and late rice under different treatments (B). Error bars represent 95% confidence interval of the mean. Different letters indicate significant differences ($p < 0.05$) between the samples. CK, no fertilizer control; NPK, mineral fertilizer; M30, 30% organic and 70% mineral fertilizer; M50, 50% organic and 50% mineral fertilizer; M70, 70% mineral and 30% mineral fertilizer.

# RESULTS

## Rice yield and soil chemical properties

The long-term fertilization significantly increased both grain and straw yields compared with CK (Fig. 1). The organic substitution for mineral fertilizers led to higher grain yield than mineral fertilizer alone (Fig. 1A). In contrast, little difference in straw yield was observed among treatments except for CK (Fig. 1B). Fertilization resulted in significant changes in soil chemical properties (Table S2). Compared with CK (pH 5.60), mineral fertilization (NPK, pH 5.06) and low proportions of organic substitution (M30, pH 5.35) decreased soil pH levels. Conversely, a high proportion of organic substitution (M70, pH 5.93) caused a significant increase in soil pH. Compared with CK, mineral fertilization and organic substitution treatments increased soil fertility and soil organic (SOC) content except for total potassium (TK). In addition, organic substitution treatments, especially a high proportion of organic addition, appeared to be superior to mineral fertilization alone in increasing soil fertility and SOC content (Table S2).

## CH$_4$ emission dynamics observed through microcosm experiments

The dynamic change of CH$_4$ emission flux during the microcosm experiment showed that CH$_4$ emissions were not prominent in the early stages of cultivation, with the peak occurring on the 18th and 23rd days (Fig. 2A). Significant differences were observed in the cumulative CH$_4$ emissions following flooding culture across various treatments, with cumulative emissions ranking as M50 > M30 > M70 > NPK > CK (Fig. 2B). Cumulative methane emissions peaked in the M50 group, reaching 10.21 μg/kg. Compared with mineral fertilization, treatment with a medium proportion of organic substitution significantly

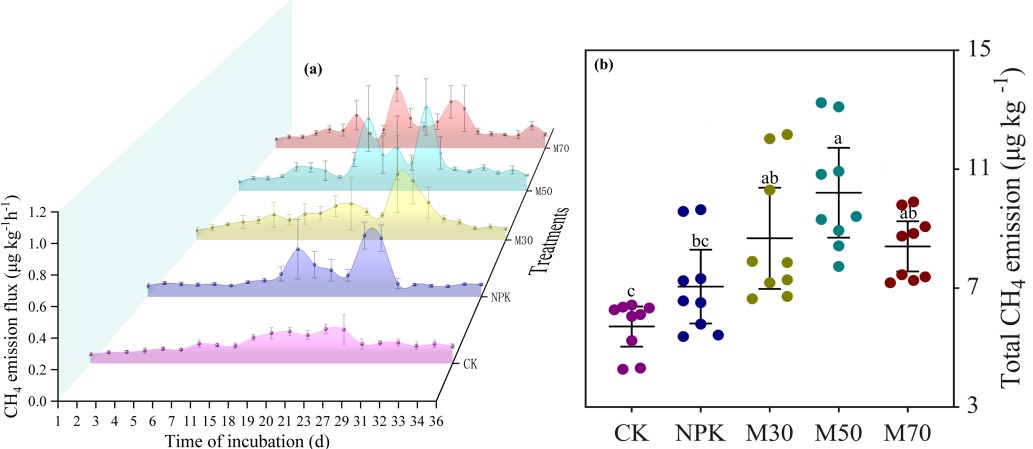

**Figure 2** **The CH$_4$ emission flux among different treatments.** The CH$_4$ emission flux with days of incubation (A) and total methane emissions (B) under different fertilization treatments. Error bars represent 95% confidence interval of the mean (B). Different letters indicate significant differences ($p < 0.05$) between the samples. CK, no fertilizer control; NPK, mineral fertilizer; M30, 30% organic and 70% mineral fertilizer; M50, 50% organic and 50% mineral fertilizer; M70, 70% mineral and 30% mineral fertilizer.

increased cumulative CH$_4$ emissions, whereas treatments with low and high proportions of organic substitutions had minimal effects (Fig. 2B).

## Abundance of methanogens and methanotrophs

Compared with the CK group, both mineral fertilization and organic substitution treatments significantly increased the gene copy numbers (*i.e.,* abundance) of *mcrA* and *pmoA* genes. For methanogens, the gene copy number of *mcrA* was the highest in the M70 group (5,520 copies/g), followed by the M50 group (4,123 copies/g) and the M30 group (2,965 copies/g). For methanotrophs, treatments with a high proportion of organic substitution exhibited an even higher abundance than those with low and medium proportions. The gene copy number of *pmoA* was the highest in the M70 group (26,672 copies/g), followed by the M50 group (6,418 copies/g) and the M30 group (3,119 copies/g) (Fig. 3). The Pearson correlation analysis revealed a significant positive correlation between the abundance of *mcrA* and the cumulative methane emission flux. Furthermore, the abundances of *mcrA* and *pmoA* showed significant positive correlations with soil pH, SOC, total nitrogen (TN), total potassium (TP), available nitrogen (AN), available phosphorus (AP), and available potassium (AK), whereas they displayed negative correlations with TK (Table S3).

## Alpha diversity analysis of methanogens and methanotrophs

A total of 3,699,882 *mcrA* sequences were obtained across all samples, ranging from 52,767 to 119,523 sequences per sample. Additionally, 2,631,826 *pmoA* sequences were obtained, with per-sample counts ranging from 30,462 to 83,877 sequences. Further, 3,912 and 1,799 methanogenic and methanotrophic OTUs were detected from 45 samples, respectively. The organic substitution treatments (M30, M50, and M70) increased the

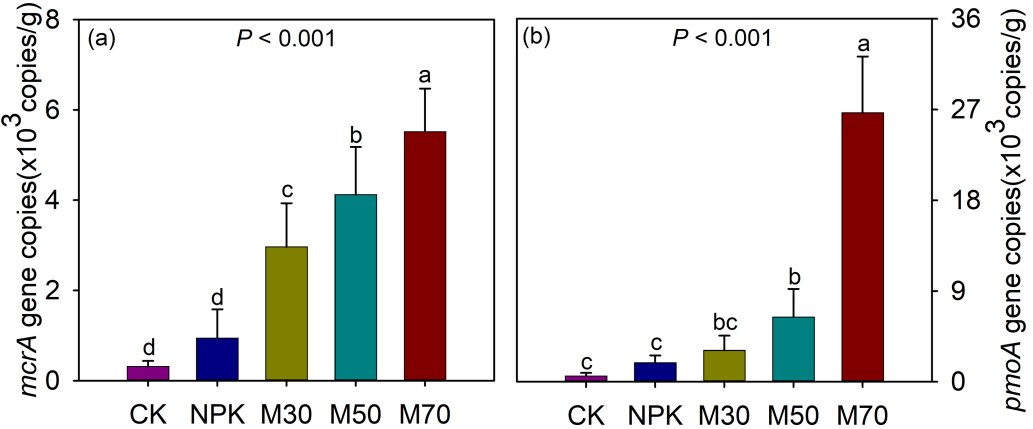

**Figure 3** **Abundance of *mcrA* (A) and *pmoA* (B) genes among different fertilization treatments.** The error bars represent the standard deviation of the mean ($n = 9$). Different letters indicate significant differences ($p < 0.05$) between the samples. CK, no fertilizer control; NPK, mineral fertilizer; M30, 30% organic and 70% mineral fertilizer; M50, 50% organic and 50% mineral fertilizer; M70, 70% mineral and 30% mineral fertilizer.

OTU richness of methanogens compared with CK (Fig. 4A). Both mineral fertilization and organic substitution treatments increased the Shannon index of methanogens but had little effect on their evenness compared with CK (Figs. 4B and 4C). Mineral fertilization and organic substitution treatments showed little effect on the OTU richness of methanotrophs compared with CK (Fig. 4D). Mineral fertilization (NPK) decreased the Shannon index and evenness of methanotrophs. In contrast, treatment with organic substitution maintained similar levels of Shannon index and evenness of methanotrophs as CK (Figs. 4E and 4F). The OTU richness of methanogens was significantly positively correlated with SOC, TN, TP, AN, and AP, and negatively correlated with TK. The methanogenic evenness was significantly negatively correlated with SOC, TN, and AN. For methanotrophs, the OTU richness was significantly positively correlated with TK and negatively correlated with AN and AK. The Shannon index of methanotrophs was significantly positively correlated with pH and negatively correlated with AK. The methanotrophic evenness was significantly positively correlated with pH, SOC, TN, TP, and AP (Table S4).

## Community composition of methanogens and methanotrophs

Fertilization significantly altered soil methanogenic and methanotrophic community compositions compared with CK. Treatment with organic substitution, especially at high proportions, exhibited a significant effect on these communities than mineral fertilization alone (Fig. 5 and Table S5). Soil pH and SOC were primary factors affecting both methanogenic and methanotrophic community compositions (Fig. 5). For methanogens, fertilization decreased the relative abundance of *Methanocella paludicola* and increased the relative abundance of *Methanobacterium oryzae* and *Methanolinea mesophile*. Treatment with low and medium proportions of organic substitution decreased the relative abundance of *Methanospirillum psychrodurum* and *Methanoperedens nitroreducens*. Treatment with organic substitution increased the relative abundance of *Methanobacterium aggregans* and

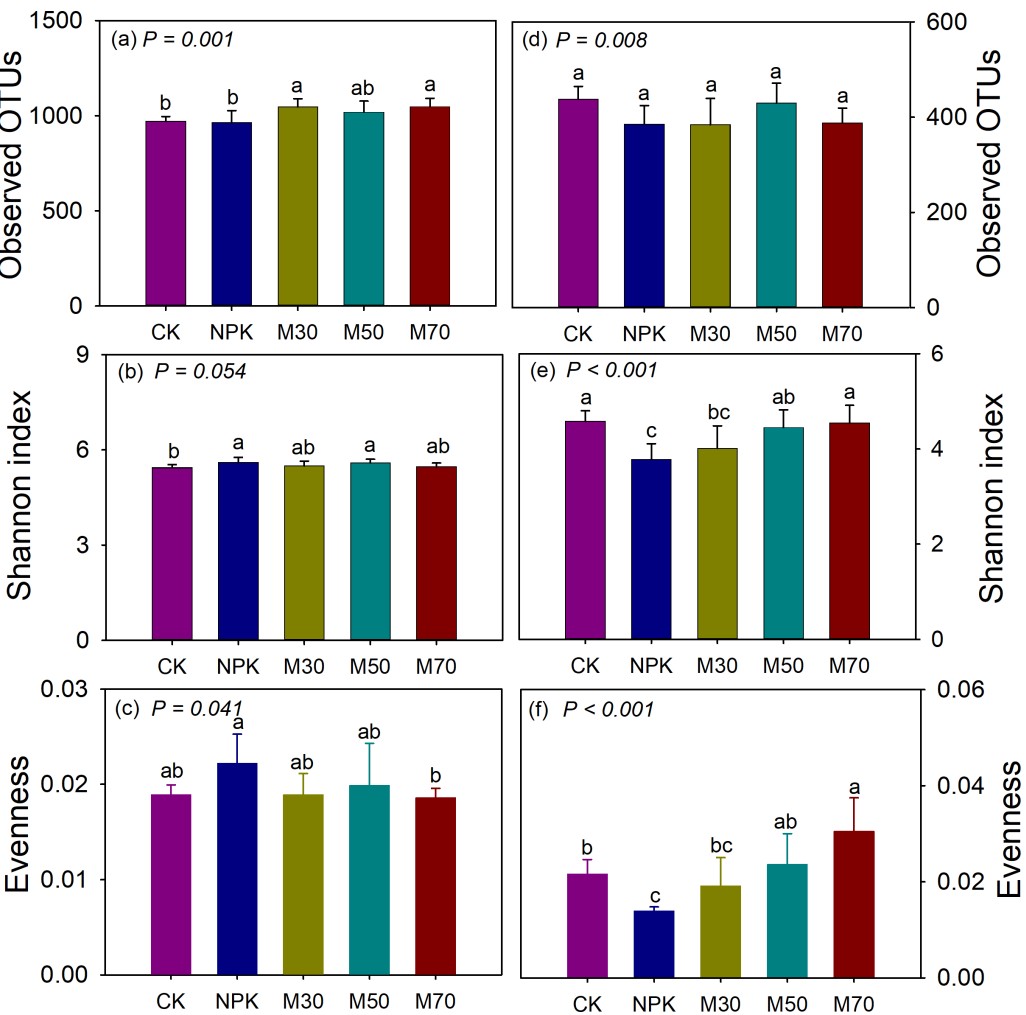

**Figure 4  Soil methanogenic and methanotrophic alpha diversity.** The methanogenic observed OTU richness (A), Shannon index (B) and evenness (C). The methanotrophic observed OTU richness (D), Shannon index (E) and evenness (F). The error bars indicate the standard error of the mean ($n = 9$). Different letters indicate significant differences ($p < 0.05$) between the samples. CK, no fertilizer control; NPK, mineral fertilizer; M30, 30% organic and 70% mineral fertilizer; M50, 50% organic and 50% mineral fertilizer; M70, 70% mineral and 30% mineral fertilizer.

*Methanobacterium bryantii*, whereas it decreased the relative abundance of *Methanosarcina thermophila* (Fig. S1).

The relative abundance of Type I and Type II methanotrophs tended to increase and decrease along organic substitution gradients, respectively (Fig. S2). The relative abundance of *Methylomonas methanica* increased and *Methylocystis bryophila* decreased under NPK fertilization and organic substitution treatments compared with CK. NPK fertilization resulted in the highest relative abundance of *Methylocystis echinoides* and *Methylocystis parvus*, and the lowest relative abundance of NC10, *Methylomarinovum caldicuralii*, and *Methylococcus capsulatus* compared with other treatments. The relative abundance of

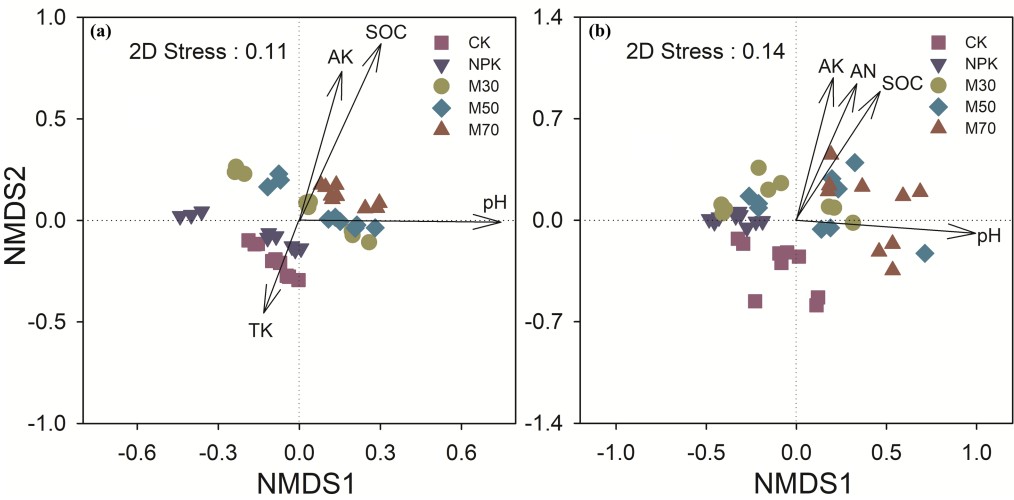

**Figure 5** Non-metric multidimensional scaling (NMDS) showing soil methanogenic (A) and methanotrophic (B) community compositions, with soil property vectors generated using envfit. The ordinations are based on Bray-curtis dissimilarities. CK, no fertilizer control; NPK, mineral fertilizer; M30, 30% organic and 70% mineral fertilizer; M50, 50% organic and 50% mineral fertilizer; M70, 70% mineral and 30% mineral fertilizer.

*Methylogaea oryzae* increased with a higher proportion of organic fertilizer substitution treatment (Fig. S3).

## DISCUSSION

The excessive use of mineral fertilizers in agro-ecosystems has recently become a common practice to sustain maximum crop yields. However, excessive application of these fertilizers can cause serious environmental pollution (*Ju et al., 2007*). Exploring new practices that reduce the use of mineral fertilizers without compromising crop yield is essential. Organic fertilizers may be a good substitute for mineral fertilizers, as the release of nutrients from organic fertilizers is more sustainable (*Liu et al., 2020*). However, previous studies have argued that organic fertilizers alone might decrease crop yield, as the nutrient release is too slow to meet the crop demand (*Kim et al., 2014*; *Thomas et al., 2019*). Therefore, this study aimed to evaluate the effect of different proportions of organic substitution for mineral fertilizers on agro-ecosystems. The findings revealed that organic substitution for mineral fertilizers significantly increased crop yield because of the effective improvement in the soil nutrient levels (*Xia et al., 2018*). Consistent with other studies, mineral fertilizers caused soil acidification (*Cai et al., 2015*). Soil acidification has been a major issue in agro-ecosystems following mineral fertilization as it results in soil degradation (*Xu et al., 2024*). However, organic substitution for sectional mineral fertilizers could offset soil acidification induced by mineral fertilizers, with a high proportion of organic substitution even increasing soil pH (*Frac, Sas-Paszt & Sitarek, 2023*). In terms of soil nutrients, crop yield, and soil health, organic substitution for partial mineral fertilizers might be an appropriate practice for agro-ecosystems.

Rice paddies are a major source of methane emissions, contributing significantly to global warming. Evaluating the effects of organic substitution for partial mineral fertilizers on methane-associated microbial communities might provide comprehensive insights into this new agricultural practice. The application of organic fertilizers increased methanogenic OTU richness and abundance compared with NPK (Figs. 3A and 4A). Some studies have shown that organic additions improve soil organic matter, providing sufficient nutrients and substrates for methanogens, thus stimulating the growth and reproduction of methanogens in rice fields (Ma et al., 2023; Wu et al., 2022; Zhang et al., 2018). The higher methanotrophic Shannon index, evenness, and abundance were observed following organic amendment (Figs. 4D and 4F). Environments with high $CH_4$ concentrations could stimulate the growth of methanotrophs (Mohanty et al., 2016), and higher substrate availability reduced competition within methanotroph taxa.

The organic substitution changed the community compositions of both methanogens and methanotrophs (Fig. 5 and Fig. S2). Wang et al. (2020) found that the application of long-term chicken manure changed the community composition of methane-associated microorganisms in paddy fields. The co-incorporation of Chinese milk vetch (*A. sinicus* L.) and rice (*Oryza sativa* L.) straw minimized $CH_4$ emissions by changing the methanogenic and methanotrophic communities in rice soils (Zhou et al., 2020b). These findings confirmed the effects of organic amendments on soil methane-associated microbial communities. Greater input of organic matter was associated with larger shifts in community composition, highlighting the critical role of soil organic matter levels on methane-associated microbial communities. As soil methanogens and methanotrophs are heterotrophic, the input of organic matter increased carbon resources and substrates available to these microbes (Zhou et al., 2020b). In addition, soil pH explained the largest variations in the community compositions of methanogens and methanotrophs (Wagner et al., 2017). Previous studies have demonstrated that soil pH affected methanogenic and methanotrophic community compositions (Wen et al., 2017). Methanotrophs, in particular, preferred a neutral soil environment, whereas acidic soil inhibited their growth, directly decreasing their abundance and methane oxidation activity (Yao, Wang & Hu, 2023).

The microcosm experiment found that treatment with a medium proportion of organic substitution (M50) increased the cumulative emission of $CH_4$ compared with mineral fertilization. However, treatment with low (M30) and high (M70) proportions of organic substitution showed little difference in cumulative $CH_4$ emissions compared with mineral fertilization (Fig. 2B). The cumulative emission of $CH_4$ depended on two aspects: $CH_4$ produced by methanogens and $CH_4$ decomposed by methanotrophs. In this study, organic amendments stimulated $CH_4$ generation by increasing *mcrA* gene abundance, with *mcrA* gene abundance increasing proportionally to the amount of organic matter input. A previous study found that the application of manure increased the gene copy number of *mcrA* (Zhang et al., 2018), which is consistent with this study. However, $CH_4$ oxidation also increased following organic addition, as evidenced by the higher abundance of the *pmoA* gene (Fig. 3B). Notably, treatment with a high proportion of organic substitution (M70) substantially increased *pmoA* gene abundance compared with other treatments.

Previous studies showed that acidic soil inhibited methanotrophic abundance. The high organic amendment increased soil pH, thus facilitating the growth of methanotrophs and enhancing their activity, leading to increased $CH_4$ oxidation (*Yoon et al., 2022*).

In addition, organic amendment increased the relative abundance of Type I methanotrophs and decreased the relative abundance of Type II methanotrophs. However, a previous study found that chicken manure amendments suppressed Type I methanotrophs in rice paddies (*Wang et al., 2020*). It typically applied chicken manure on top of mineral fertilizers, which differed from this study, where mineral fertilizers were partially replaced with organic alternatives. This difference in methodology may have led to varying results. Previous studies showed that increasing the soil nitrogen content benefited the growth of Type I methanotrophs (*Lee et al., 2009*). In this study, organic amendment enriched the soil with higher nitrogen content (Table S2), further facilitating the growth of Type I methanotrophs.

Specifically, Type I methanotrophs prefer environments with high methane concentrations and low oxygen concentrations, whereas Type II methanotrophs prefer environments with low methane concentrations and high oxygen concentrations (*Bull et al., 2000*). The organic amendment, especially the high organic application, substantially changed the composition of the methanotrophic community by increasing the ratio of Type I/Type II methanotrophs, thereby enhancing the $CH_4$ oxidation (*Ma, Conrad & Lu, 2013*; *Shrestha et al., 2008*).

Based on the aforementioned results, this study proposed a $CH_4$ generation and oxidation process following organic substitution. Treatment with a low proportion of organic amendment (M30) caused the input of organic matter to increase soil carbon resources and stimulate the growth of soil microorganisms. The limited carbon source was mainly used by soil microorganisms, resulting in relatively weak stimulation of $CH_4$ production and oxidation processes. This led to similar cumulative $CH_4$ emissions comparable to those observed with mineral fertilization treatment. Compared with a low proportion of organic substitution, the higher carbon input with a medium proportion of organic substitution (M50) significantly stimulated the *mcrA* abundance (*Kim et al., 2014*). However, treatment with a medium proportion of organic substitution had little effect on *pmoA* abundance, possibly due to the relatively acidic soil environment (Fig. 3B) (*Wang et al., 2018*), resulting in higher cumulative $CH_4$ emissions. Although a high proportion of organic amendment (M70) increased *mcrA* abundance, it also increased soil pH, triggering a huge *pmoA* abundance and enhancing the relative abundance of Type I methanotrophs, significantly facilitating $CH_4$ oxidation. This resulted in relatively low cumulative $CH_4$ emissions in M70.

This study had certain limitations. It is essential to explore the underlying mechanisms behind shifts in methane-associated microbial community compositions following organic substitution and understand how specific organic amendments alter microbial interactions and metabolic pathways. In addition, although organic fertilizers play a significant role in soil microbial communities, predicting the optimal ratio of organic substitution for mineral fertilizers remains challenging, resulting in incomplete research on soil methane-related microorganisms. These limitations should be addressed in future research.

## CONCLUSIONS

In this study, the organic substitution of mineral fertilizer improved soil fertility, alleviated soil acidification, increased crop yield, enhanced methanogenic and methanotrophic abundance, and changed their community structures. However, treatment with a medium proportion of organic substitution increased cumulative $CH_4$ emissions, which might contribute to global warming. Treatment with a high proportion of organic substitution optimized the methanotrophic community composition by increasing the relative abundance of Type I taxa. Besides, high organic substitution treatment increased soil pH, resulting in a higher *pmoA* abundance. Thus, treatment with a high proportion of organic substitution enhanced methane oxidation capacity without increasing cumulative $CH_4$ emissions compared with mineral fertilizer alone. In addition, this treatment increased crop yield and reduced the reliance on mineral fertilizers, thereby minimizing environmental pollution. Overall, replacing chemical fertilizers with a high proportion of organic matter might be a sustainable agricultural practice in paddy ecosystems.

### Funding

This study was financially supported by the National Key Research and Development Program of China (2021YFD1700203), the National Natural Science Foundation of China (NO. 42267046) and the Outstanding Youth Research Project of Anhui Province for Xingjia Xiang (2022AH030015), China Agriculture Research System of MOF and MARA (No. CARS-22). The funders had no role in study design, data collection and analysis, decision to publish, or preparation of the manuscript.

### Grant Disclosures

The following grant information was disclosed by the authors:
The National Key Research and Development Program of China: 2021YFD1700203.
The National Natural Science Foundation of China: 42267046.
The Outstanding Youth Research Project of Anhui Province: 2022AH030015.
China Agriculture Research System of MOF and MARA: CARS-22.

### Competing Interests

The authors declare there are no competing interests.

### Author Contributions

- Dandan Yuan conceived and designed the experiments, performed the experiments, analyzed the data, prepared figures and/or tables, and approved the final draft.
- Keke Dang analyzed the data, authored or reviewed drafts of the article, and approved the final draft.
- Jing Yin analyzed the data, prepared figures and/or tables, and approved the final draft.
- Han Liu performed the experiments, analyzed the data, prepared figures and/or tables, and approved the final draft.

- Tingting Ma performed the experiments, analyzed the data, prepared figures and/or tables, and approved the final draft.
- Jia Liu conceived and designed the experiments, authored or reviewed drafts of the article, and approved the final draft.
- Xingjia Xiang conceived and designed the experiments, analyzed the data, authored or reviewed drafts of the article, and approved the final draft.

## Data Availability

The raw data are available at NCBI SRA: PRJNA1116972, SAMN41388671–SAMN41388675 and SAMN41553632–SAMN41553636.

## Supplemental Information

Supplemental information for this article can be found online at http://dx.doi.org/10.7717/peerj.19000#supplemental-information.

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
