# Peer review of "Effects of different proportions of organic substitution for mineral fertilizers on soil methanogenic and methanotrophic communities in paddy fields"

_PeerJ, doi:10.7717/peerj.19000_

## Round 0.1 · original submission · Major Revisions

- Provide detailed methodologies and conduct additional experiments to strengthen the reliability and robustness of the results.

- Emphasize the significance of the findings, explaining their importance and how this work differs from prior studies in the field.

- Cite relevant references to ensure the work is well-supported and contextualized within existing literature

- The abstract is too qualitative. Incorporating quantitative analysis would better highlight the improvements achieved in this study.

- The manuscript would be enhanced by the inclusion of more comprehensive quantitative data to substantiate the findings. Specifically, providing detailed values for methane (CH₄) emissions, microbial abundances, and soil pH changes across various treatment groups would increase the transparency and robustness of the results. To facilitate a thorough analysis and ensure reproducibility, it is recommended that the authors provide raw read data from the bioinformatics analyses.

- The manuscript mentions soil pH changes but could benefit from a more detailed examination of other influential environmental factors, such as temperature and moisture. Analyzing how these factors interact with the organic treatments to influence CH₄ emissions would bolster the robustness of the conclusions.

Reviewer 1 ·

Basic reporting

1. A brief explanation of what the mcrA and pmoA genes are and their respective roles in the carbon cycle—mcrA in methane production and pmoA in methane oxidation.
2. Discussion on the significance of these genes in soil microbial communities, particularly to agricultural practices and their impact on greenhouse gas emissions.

Experimental design

The primer sequences should be provided to ensure reproducibility and facilitate the validation of experimental methods by other researchers.

Validity of the findings

1. While the study identifies the effects of different proportions of organic substitution on methane emissions and microbial communities, it may not sufficiently explore these changes' underlying mechanisms. A deeper investigation into how specific organic amendments influence microbial interactions and metabolic pathways could strengthen the findings.
2. The manuscript mentions changes in soil pH due to organic amendments. Still, it may not adequately address how other environmental factors (e.g., temperature, and moisture) interact with these treatments to influence methane emissions. A more thorough examination of these variables could enhance the robustness of the conclusions.
3. While the study compares different organic substitution treatments, it may not sufficiently discuss how these results compare to existing literature or previous studies. A more thorough comparison could provide context and highlight the significance of the findings.

Additional comments

1. The results may report changes in microbial community composition and methane emissions but might not delve into the mechanistic insights behind these changes. Discussing how specific treatments influenced microbial interactions or metabolic pathways would provide a deeper understanding of the results.
2. Including references to previous research that utilized these genes to explore methane dynamics in different ecosystems would strengthen the study's rationale and position it within the broader scientific literature.
3. There may be instances of unclear phrasing or complex language that could hinder comprehension for a broader audience. Improving the clarity of the writing and ensuring that technical terms are well-defined would enhance the manuscript's accessibility.

Reviewer 2 ·

Basic reporting

1. The standard of English is good.
2. Some of the references do not seem directly relevant to the point being made in the article (I have noted where in the manuscript).
3. Figures should show points rather than bars, and 95% Confidence Intervals (+ and -) rather than SE. The raw data has been shared.
4. The manuscript is self-contained with results relevant to hypotheses.

Experimental design

1. The research is 'original' only in that these specific results haven't been published previously. There are dozens of similar papers in the literature already, covering the effects of manure fertilizers on methanogens. In this sense, the paper is not original.
2. Given the large number of similar studies already published, the research does not fill a knowledge gap.
3. The research appears to be technically sound. It wasn't clear why an OTU separation of 93% was used rather than 97%.
4. It wasn't clear how the 45 samples (9 reps, 5 treatments) were physically established (45 separate plots?).

Validity of the findings

1. The paper replicates many similar studies.
2. Data to replicate the figures has been provided. Raw sequence data accession numbers are provided.
3. The conclusions are well stated but also well known (e.g. that substitution of mineral fertilizer with manure raises soil pH).

Additional comments

No comment.

Annotated reviews are not available for download in order to protect the identity of reviewers who chose to remain anonymous.

---

## Round 0.2 · Minor Revisions

There are some minor comments that require the authors to improve the explanation for clarity.

Additionally, please kindly check the English.

Reviewer 1 ·

Basic reporting

The authors have improved the efficiency and clarity of the English language used in the manuscript.

Experimental design

The authors have added the details into the Materials and Methods section as suggested.

Validity of the findings

All suggestions have been addressed and corrected by the authors.

Reviewer 2 ·

Basic reporting

Good

Experimental design

Good

Validity of the findings

Good

Additional comments

The authors have addressed me previous concerns and I am happy to recommend publication of this paper.

In the Abstract, please correct the sentence "Treatment with organic substitution signiûcantly changing their community composition."
Also "Treatment with high proportion of organic substitution enhenced"
There may be other spelling/grammatical mistakes in the manuscript which I have missed, please check carefully.

---

## Round 0.3 · accepted · Accept

The manuscript has been significantly improved, addressing the key points raised by the reviewers. This version meets the quality standards required for publication.